# Cervicovaginal-Microbiome Analysis by 16S Sequencing and Real-Time PCR in Patients from Novosibirsk (Russia) with Cervical Lesions and Several Years after Cancer Treatment

**DOI:** 10.3390/diagnostics13010140

**Published:** 2023-01-01

**Authors:** Mikhail K. Ivanov, Evgeny V. Brenner, Anastasia A. Hodkevich, Victoria V. Dzyubenko, Sergey E. Krasilnikov, Alphiya S. Mansurova, Irina E. Vakhturova, Eduard F. Agletdinov, Anastasia O. Shumeikina, Alyona L. Chernyshova, Sergei E. Titov

**Affiliations:** 1Department of the Structure and Function of Chromosomes, Institute of Molecular and Cellular Biology, Novosibirsk 630090, Russia; 2AO Vector-Best, Novosibirsk 630117, Russia; 3Department of Natural Sciences, Novosibirsk State University, Novosibirsk 630090, Russia; 4Institute of Oncology and Neurosurgery, E. Meshalkin National Medical Research Center, Novosibirsk 630055, Russia; 5Hospital (Medsanchast) No. 168, Novosibirsk 630117, Russia; 6Cancer Research Institute, Tomsk National Research Medical Center of the Academy of Sciences, Tomsk 634009, Russia

**Keywords:** high-grade squamous intraepithelial lesion, cervical cancer, cancer treatment, cervicovaginal microbiome, 16S rRNA gene sequencing, post-treatment effect, *Lactobacillus iners*, *Cutibacterium acnes*, real-time PCR, human papillomavirus

## Abstract

Disturbed cervicovaginal-microbiome (CVM) structure promotes human papillomavirus (HPV) persistence and reflects risks of cervical lesions and cancer onset and recurrence. Therefore, microbiomic biomarkers may be useful for cervical disease screening and patient management. Here, by 16S rRNA gene sequencing and commercial PCR-based diagnostic kits, we profiled CVM in cytological preparations from 140 HPV-tested women (from Novosibirsk, Russia) with normal cytological findings, cervical lesions, or cancer and from 101 women who had recently received different cancer therapies. An increase in lesion severity was accompanied by higher HPV prevalence and elevated CVM biodiversity. Post-treatment CVM was found to be enriched with well-known microbial biomarkers of dysbiosis, just as in cervical disease. Nonetheless, concentrations of some skin-borne and environmental species (which gradually increased with increasing lesion severity)—especially *Cutibacterium* spp., *Achromobacter* spp., and *Ralstonia pickettii*—was low in post-treatment patients and depended on treatment types. Frequency of *Lactobacillus iners* dominance was high in all groups and depended on treatment types in post-treatment patients. Microbiome analysis via PCR-based kits revealed statistically significant differences among all groups of patients. Thus, microbiome profiling may help to find diagnostic and prognostic markers for management of cervical lesions; quantitative PCR-based kits may be suitable for these purposes.

## 1. Introduction

Precancerous cervical lesions and cervical cancers are generally accompanied by changes in the cervicovaginal microbiome (CVM). Normally, in women of reproductive age, the CVM is dominated by lactobacilli (LBs), which maintain a weakly acidic pH (≤4.5; due to the synthesis of lactic acid from glycogen); produce lysozyme, hydrogen peroxide, and bacteriocins; prevent biofilm formation; and have a stimulatory influence on local immunity. All this taken together prevents colonization by bacterial and viral pathogens, including high-risk human papilloma virus (HR-HPV) [1,2,3]. If the concentration of LBs decreases, then other, less protective or unprotective microorganisms take their place, resulting in dysbiosis, growing biodiversity, and an elevated risk of infections and pathological conditions. This situation is usually a consequence of weakening of either systemic or local immunity [4,5,6].

The risk of formation and progression of pre-cancerous cervical lesions is undoubtedly related to the CVM because the latter modulates local immunity, and its dysbiotic states raise the risk of HR-HPV infection and persistence, which in turn trigger squamous intraepithelial lesion (SIL) formation [7,8]. On the other hand, HR-HPV infection itself is a widespread condition, and only a small proportion of infected individuals develops lesions; this state of affairs reduces the positive predictive value of HPV testing [9,10]. Therefore, although HR-HPV detection is currently the key cervical screening method, there is a problem of stratification of HPV-positive patients by lesion risk. This problem is far from solved via the leading existing “co-testing” approach, when HPV testing is combined with cervical cytological analysis, because the latter has many limitations and strongly depends on human factors [11,12]. In this regard, it appears reasonable to combine HPV testing with molecular analysis of CVM features (performed on the same clinical specimen within the same pipeline) during management of a cervical disease in order to increase the positive predictive value. The “gold standard” for such analysis is next-generation sequencing (most often 16S rRNA gene sequencing), which, however, is relatively expensive and low-throughput. We propose that cervicovaginal-microbial-community profiles related to an elevated cervical lesion risk can be diagnosed by much simpler and cheaper quantitative PCR (qPCR)-based tools targeting molecular markers derived from next-generation sequencing-based studies, as is the case for commercially available qPCR-based kits aimed at diagnosing bacterial vaginosis (BV) and aerobic vaginitis (AV) [13,14,15]. Moreover, as microbiomic markers of cervical disease, BV and AV significantly overlap [7,16], and one can assume that the kits that are employed to detect BV and AV may be helpful for cervical disease screening and patient management “as is”. Another possible application of such kits is post-treatment monitoring. Information about CVM alterations after cancer treatment, in particular about its long-term effects for different treatment types, is very limited. We can speculate that the post-treatment state of the cervicovaginal microbial community may affect the risk of recurrence, and patients may benefit from the analysis of cervicovaginal-microbial-community structure by molecular methods during post-treatment monitoring. Nonetheless, for the correct interpretation of such data, regional features of microbial communities may be important, and knowledge about them is currently incomplete [17,18].

In this regard, to find suitable microbiomic biomarkers of cervical disease and of post-treatment restoration and to assess the possible usefulness of an existing PCR-based diagnostic tool for this purpose, we performed microbiome analysis by two methods (16S rRNA gene sequencing and qPCR-based commercial diagnostic kits used to diagnose dysbioses in women) on cervical cytology preparations from HPV-tested women from Novosibirsk (Russia) with different levels of cervical damage, as well as on cervical cytology preparations from women after different types of cervical cancer treatment.

## 2. Materials and Methods

### 2.1. Clinical Material and Nucleic-Acid Extraction

The cytological material was obtained in accordance with the laws and regulations of the Russian Federation. All patients provided informed consent, and the clinical data were depersonalized. Air-dried cervical epithelial swabs were prepared by routine Papanicolaou staining. The specimens were collected from patients undergoing cytological and histological examination and treatment in oncogynecology departments of Novosibirsk Regional Oncology Dispensary and Hospital (Medsanchast) No. 168 (Novosibirsk, Russia). The clinical data were retrieved from databases of the respective medical centers. The samples were classified according to the Bethesda system: normal cytology (negative for intraepithelial lesion or malignancy; NILM), low- and high-grade squamous intraepithelial lesions (LSILs and HSILs), or invasive cervical cancer (CC). All HSIL and CC diagnoses and some LSIL diagnoses were verified histologically. Additionally, cytological preparations from patients previously treated in Novosibirsk Regional Oncology Dispensary for CC were included in the study. This subsample included patients who underwent treatment 2–7 years (median 2.7 years) before the specimen collection and had no signs of disease recurrence. All patients enrolled in the study were nonpregnant, and did not undergo hormonal treatment, antibiotic treatment, or probiotic correction of cervicovaginal microflora during the treatment.

Total-nucleic-acid extraction from the cytology preparations was performed as described elsewhere [19]. Primers and TaqMan probes for the human *HMBS* gene (111 bp) and for a conserved region of the bacterial 16S rRNA gene (302 bp) were used to verify DNA integrity. Specimens with an insufficient human DNA amount (C_q_ for *HMBS* > 35) or bacterial DNA amount (C_q_ for the 16S gene fragment > 32) were excluded from further analyses. The final study population was composed of specimens from 241 women: NILMs (n = 77; patients’ mean age: 39.3 years), LSILs (n = 24; mean age 40.1 years), HSILs (n = 22; mean age 44.5 years), CCs (n = 17; mean age 51 years), and post-treatment (PT) patients (n = 101; mean age 52.4 years). The PT group included patients after chemo-radiotherapy (CRT) (n = 26; mean age 49.5 years), radiation therapy (RT) (n = 46; mean age 55.1 years), surgical treatment (ST) (n = 18, mean age 51.0 years), or a combination of surgical treatment and RT (hereafter: combined therapy, CT) (n = 11; mean age 48.3 years). Subgroups of patients after different treatments did not differ statistically significantly in age, nor were age differences identified among cancer stage (I, II, and III) subgroups. Surgical treatment was prescribed only to patients with stage I, and RT was prescribed at stage ≥ II.

### 2.2. PCR-Based Analyses

All amplification procedures were performed using a CFX96 thermal cycler (Bio-Rad Laboratories, Hercules, CA, USA) and qPCR-based kits produced by AO Vector-Best (Novosibirsk, Russia). To determine relative enrichment with LBs and the proportion of typical opportunistic species in the total bacterial population, the RealBest BioFlor Kit was utilized. The kit targets *Lactobacillus* spp., *Gardnerella vaginalis*, *Atopobium vaginae*, *Prevotella* spp., *Leptotrichia amnionii* group, *Streptococcus* spp., *Staphylococcus* spp., and *Enterococcus* spp., and uses the 16S rRNA gene as the internal reference gene. Identification of *Mobiluncus mulieris, Mobiluncus curtisii, Saccharimonas aalborgensis* (TM7)*,* and *BVAB2* DNA was carried out using RealBest Mobiluncus mulieris/Mobiluncus curtisii DNA and RealBest Saccharimonas aalborgensis (TM7)/BVAB2 DNA Kits, respectively. Identification, genotyping, and quantification of HR-HPV viral DNA were performed by means of the RealBest HPV Genotype, Quantitative Kit, which targets the 12 most prevalent HR-HPV types (HPV 16, 18, 31, 33, 35, 39, 45, 51, 52, 56, 58, and 59) and determines type-specific viral loads per 100,000 cells. One CC sample, initially identified as HPV-negative (see the Results Section), was additionally tested for HPV 26, 53, 66, 68, 73, and 82 with RealBest DNA HPV 26/53/66 and RealBest DNA HPV 68/73/82 Kits. RealBest BioFlor, RealBest Mobiluncus mulieris/Mobiluncus curtisii DNA, and RealBest Saccharimonas aalborgensis (TM7)/BVAB2 DNA kits target different taxon-specific regions of the 16S rRNA gene. All kits used for HPV genotyping target genotype-specific regions of the HPV *E6* gene.

### 2.3. Preparation of a 16S rRNA Amplicon Library

Dual-indexed libraries were prepared by two-round PCR. Primers Fus342F and Fus806R flanking v3 and v4 hypervariable regions (~456 bp) of the 16S rRNA gene and bearing universal adapter sequences for the subsequent 2nd PCR round on the 5′ end were used for the 1st PCR round. This round was implemented with PrimeSTAR^®^ GXL DNA Polymerase (Takara Bio Inc, Kusatsu, Japan) in a 20 μL reaction mixture with a primer concentration of 0.3 μM. Thermal cycling conditions were as follows: 98 °C for 1 min followed by 4 cycles at 98 °C for 10 s, 55 °C for 15 s, and 68 °C for 40 s, followed by 18 cycles, each consisting of 98 °C for 10 s and 68 °C for 40 s.

Prior to the 2nd PCR round, the reaction mixtures were diluted 100-fold with 10 mM Tris-EDTA buffer (pH 8), and 1 μL of this diluted solution per reaction was used. This round was set up by means of indexed primers P7i and P5i designed to anneal to a universal adapter. The amplification was performed with Taq polymerase (Vector-Best) in a 20 μL reaction mixture with primer concentration of 0.3 μM. The following amplification program was executed: 94 °C for 4 min followed by 12 cycles of 98 °C for 12 s and 65 °C for 1 min. Libraries were purified on AMPure XP paramagnetic beads (Beckman Coulter, Brea, CA, USA) and quantified by TaqMan qPCR with primers annealing to adapter sequences (primers ilP-f and ilP-r and probe ilP-p).

The sequences of all oligonucleotides used in the study (except those in commercial kits) are shown in Appendix A.

### 2.4. 16S rRNA Gene Sequencing and Sequencing Data Analysis

The sequencing was performed on an Illumina MiSeq instrument. Thus, 23.2 million 2 × 250 nt PF paired-end reads were generated, resulting in 45.5 thousand PF paired-end reads per library on average. Initial quality filtration and trimming of adapters’ sequences were performed in the Trimmomatic (v.0.39) software run in “PE” mode [20]. The resulting fastq files containing high-quality reads with adapter sequences trimmed were next analyzed with the QIIME2 (v.2021.11) microbiome analysis package. Sequence quality control, deletion of primer sequences, filtering out chimeras, generating a set of representative sequences, and construction of a feature table were performed within the DADA2 pipeline of QIIME2 run in “denoise-paired” mode. A minimum of 1500 high-quality reads were analyzed, which provided sufficient sequencing depth for a sample. Taxonomic analysis was performed using the QIIME2 package with the help of a pre-trained weighted Naive Bayes taxonomic classifier based on the Silva (release 138) rRNA database. Additionally, α-diversity scores (observed features, Shannon entropy, and Faith’s phylogenetic diversity) were calculated with the help of corresponding QIIME 2 (v.2021.11) modules. For analysis outside of the QIIME2 package, the resulting feature and taxonomy tables were exported into CSV file format and analyzed in Microsoft Excel and using Python 3.9 modules. Thus, data filtration and score calculations for heatmap construction were performed with pandas and numpy modules, while heatmap visualizations were performed by means of the seaborn module.

### 2.5. Other Data Analyses

The data processing was conducted in Excel (Microsoft, USA) or STATISTICA v.9.1 (TIBCO Software Inc., USA). The Bonferroni correction was applied to adjust findings for multiple testing. An adjusted *p*-value was calculated as a single-comparison (i.e., pairwise) nonadjusted *p*-value multiplied by the total number of comparisons. Data with an adjusted *p*-value < 0.05 were considered statistically significant. LEfSe analysis and visualizations were performed by means of the Galaxy module run at https://huttenhower.sph.harvard.edu/lefse/ (accessed on 20 November 2022) [21].

## 3. Results

### 3.1. HR-HPV Detection, Genotyping, and Quantitation

HPV DNA of at least one high-risk genotype was detected in 172 of 241 (71.3%) samples: in 16 of 17 (94.1%) CCs, in all (100%) HSILs, in 19 of 24 (79.2%) LSILs, in 19 of 77 (24.6%) NILMs, and in 96 of 101 (95.0%) PT patients’ samples. In all groups, HPV16 was the most prevalent (Figure 1). Relative frequency of the most aggressive genotypes 16 and 18 (summarized) was 43.5% among NILMs, 47.7% among HSILs, 44.9% among HSILs, 48.6% among CCs, and 40.4% among PT samples. Multiple infection (2–6 genotypes) was detected in 110 of 171 HPV-positive samples (64.3%): in 10 of 16 (62.5%) CCs, in 12 of 22 (50%) HSILs, in 13 of 19 (68.4%) LSILs, in 5 of 19 (26.3%) NILMs, and in 71 of 97 (73.2%) PT samples. No additional high-risk genotypes were found in the CC specimen that initially tested HPV-negative. Although HR-HPV occurrence was significantly higher in HSIL, CC, and PT groups than in the NILM group, the viral loads of HPV-positive samples did not differ significantly among the diagnoses and treatment types.

### 3.2. CVM Analysis by 16S rRNA Gene Sequencing

#### 3.2.1. General Biodiversity

At the operational taxonomic unit (OTU) level, microbial α-biodiversity and richness in the groups was estimated using observed features, Faith’s, and Shannon’s indices. Evaluation of rarefaction curves—constructed by means of the number of observed OTUs for different values of rarefaction depth for each sequenced sample—showed that the curves tended to reach a plateau, suggesting that the obtained sequences were sufficient to cover real biodiversity. All diversity parameters increased reliably from NILM to CC. Biodiversity and richness were significantly lower in PT samples from patients undergoing surgery (PT+) (as the main method or as part of combined regimens) as compared to patients without such an intervention (PT-), where these parameters were relatively high (Figure 2). Further stratification of PT patients by treatment type did not reveal statistically significant differences in biodiversity parameters.

#### 3.2.2. Enrichment with LB Species

Relative abundance of LBs decreased from NILMs to CCs. The PT group was relatively heterogeneous in this parameter, similarly to severe lesions (Figure 3A; see also Figure 5E). The most prevalent LB species were *L. iners* and *L. crispatus*; *L. gasseri* and *L. jensenii* were less frequent; the other 13 LB species were found rarely and mainly at low concentrations. For 163 samples with ≥1000 LB-mapped reads, we estimated the relative proportion of different LB species in the total LB fraction. Enrichment and dominance of different species varied between groups. Figure 3B presents relative dominance frequencies of the most abundant LB species in our groups. The most frequent dominance of *L. iners* was characteristic of the PT group; the other groups did not differ in this parameter. The PT group itself was nonhomogeneous in the enrichment with *L. iners* and non-*iners* LB species (see Figure 7C). Unexpectedly, *L. gasseri* dominance was most typical of CC. No dependence of the biodiversity indices on the dominance of any LB species was found.

#### 3.2.3. Community State Types

According to ref. [22], human vaginal microbial communities can be classified into five community state types (CSTs). Four CSTs (I, II, III, and V) are dominated by *L. crispatus*, *L. gasseri*, *L. iners*, and *L. jensenii*, respectively, whereas CST IV (“diverse group”) is unfavorable, has no specific dominant species, and may be further subdivided into subtypes [23]. Based on a simplified criterion of predominant species for determining the frequency of different CSTs in our study population, we found a gradual decrease in the proportion of LB-dominated CSTs with increasing disease severity, and the opposite trend for CST IV. CST IV was the most prevalent in all groups except for NILM; CST II and V were minor types (Figure 3C). In the PT group, CST IV frequency was lower among PT+ cases. Nevertheless, after hierarchical clustering based on Bray–Curtis dissimilarity between all pairs of community states and an average linkage, our study population split into clusters that did not match Ravel’s classification (see Figure 4).

#### 3.2.4. Non-LB Species

In our study population, DNA of 204 species from 57 families of bacteria were found, for which at least 1% of reads were detectable in at least one sample (see Figure 4). In receiver-operating characteristic curve analysis, the percentage of non-LB-mapped reads showed a large (0.778 ± 0.670) area under the curve as a surrogate diagnostic marker of high cervical-lesion risk (Figure 5C). Overall, HSILs, cancers, and PT patients’ samples were found to be enriched with families and species typical of BV and/or AV, as compared to HSILs and NILMs, and this enrichment increased with increasing severity of lesions (Figure 5A,E, left; Appendix A). Most of these species, including well-known markers of cervical disease (such as *Prevotella* spp., *Veillonella* spp., *Megasphaera* spp., *Mobiluncus* spp., *Peptostreptococcus* spp., and *Sneathia* spp.) remained over-represented in the PT group (see an example in Figure 5A). Nonetheless, we found notable exceptions to this pattern for several obligate or facultative aerobes. Particularly, the four families were not only (a) highly abundant in all groups and (b) gradually increased in relative abundance with increasing lesion severity, but also (c) showed decreased abundance in all PT subgroups, down to values comparable to NILMs. These were Burkholderiaceae (predominantly *Ralstonia pickettii*), *Alcaligenaceae* (predominantly *Achromobacter* species), Propionibacteriaceae (predominantly *Cutibacterium acnes*), and Moraxellaceae (predominantly *Acinetobacter lwoffi*) (Figure 5B,E, right). Among these species, *C. acnes* enrichment within the PT group was most dependent on the treatment type, being minimal after CT (Figure 5D).

Aside from *C. acnes*, an enrichment increase with increasing lesion severity and its decrease in the series “CRT, RT, ST, CT” were noted for *Corynebacterium* species (23 species, mainly *C. tuberculostearicum*) and *Staphylococcus* species (four species, mainly *S. warneri*), which are also known to be abundant in skin microbial communities [24]. At the same time, *Streptococcus* species did not manifest an increase in relative abundance with lesion severity; however, among PT patients, *Streptococcus* abundance was lower if treatment included surgery. No correlation was found between relative abundance levels of any of the aforementioned species among individual samples. Other species typical of skin communities were much less abundant.

Significant differences in the microbiome were found when the linear discriminant analysis for effect size (LEfSe) algorithm was applied at all OTU levels. LEfSe identified statistically significant features (an LDA score of at least 2, Kruskal–Wallis sum-rank test, *p* < 0.05) that best characterize each group of patients, particularly 19 genera from 10 bacterial families that discriminated patients by diagnosis and 14 genera from 10 families that discriminated PT+ and PT- patients (Figure 6). Among the specific markers of the PT group, only Bifidobacteriaceae and Rhizobiaceae were highly abundant. Being dependent on the treatment type, they manifested different patterns (Figure 7A). Additionally, some much less abundant families—Campylobacteraceae (predominantly *Campylobacter ureolyticus*) and Actinomycetaceae—turned out to be much more enriched within the PT group (Figure 7B). The latter family was represented by *Mobiluncus* genus species, which, unlike other BV-associated anaerobic species and similarly to skin-borne aerobic species, not only increased in relative abundance with increasing lesion severity but also had even greater relative abundance in the PT group (*p* = 0.016), gradually decreasing in the series “CRT, RT, ST, CT.”

At the level of species, the LEfSe analysis identified *G. vaginalis* and *Prevotella* species as even better biomarkers of PT cervicovaginal microbiome than of cervical disease-associated microbial community. This finding is supported by Figure 7C, which allows one to compare the enrichment in the four biomarkers: non-iners LBs (the basis of a healthy CVM), *L. iners* (the basis of “transient” CST III), *G. vaginalis*, and *Prevotella* spp. (BV-associated markers of anaerobic dysbiosis, the main species forming biofilms and the basis of “adverse” CST IV). Relative abundance of both anaerobic species was usually excessive in the PT group (even when compared to severe lesions and cancers) and was higher in the presence of *L. iners* dominance. After stratification by treatment type, they showed different patterns.

### 3.3. CVM Analysis by PCR-Based Kits

The accuracy of quantification of LB and non-LB species’ DNA was directly compared between the RealBest BioFlor Kit and 16S rRNA gene sequencing for a subgroup of 181 specimens (some samples were excluded from the analysis for various technical reasons). The two methods overall demonstrated a high correlation; Spearman correlation coefficients for different species percentages measured by the RealBest BioFlor Kit and 16S rRNA gene sequencing ranged between 0.98 for *Enterococcus* spp. to 0.72 for *Streptococcus* spp., being 0.78 for *Lactobacillus* spp., 0.80 for *G. vaginalis*, 0.76 for *Prevotella* spp., and 0.79 for all non-LB species taken together. The most discordant results were not explainable by ineffective detection of individual species and could be due to biases introduced by the classification algorithms that were applied to the raw next-generation sequencing data and/or (more possibly) by systematic error of the PCR method.

The BioFlor Kit stratifies analyzed clinical samples by the LB DNA proportion into three categories using two cutoff points: category 1, the proportion of LB DNA > 80% (nominally healthy state); category 2, LB DNA proportion 20–80% (moderate dysbiosis); and category 3, LB DNA proportion 0–20% (pronounced dysbiosis). As depicted in Figure 8A, this categorization clearly reflects lesion severity but is unacceptable as a diagnostic criterion of the post-treatment state. In receiver-operating characteristic curve analysis, the percentage of non-LB 16S rDNA copies as assessed by the BioFlor Kit was almost identical to that determined by 16S gene sequencing (area under the curve 0.771 ± 0.47 for this parameter as a surrogate diagnostic marker of high cervical lesion risk; Figure 8B, see also Figure 5C).

Relative abundance of LB and of selected BV- and AV-associated opportunistic species for different patient groups as determined by the BioFlor Kit is displayed in Figure 8C and clearly reflects differences in the CVM state among all the analyzed groups.

In Figure 8D, readers can see that the detection frequency of most species detected by the BioFlor Kit increased with increasing lesion severity and was still high in PT patients. This trend was even more pronounced for opportunistic species *Mobiluncus mulieris*, *M. curtisii*, TM7, and BVAB2, whose probability of detection by RealBest kits also clearly correlated with lesion severity. All the above-mentioned species with high prevalence were detectable in the PT group regardless of treatment types.

## 4. Discussion

The factors determining whether HR-HPV will persist or disappear after initial infection are diverse, and causal relations between most of them are ambiguous. This state of affairs hinders the application of these parameters to the personalized management of HPV-positive patients and their stratification by risks of SIL and CC. So far, the list of tools for such stratification is short. Currently, so-called partial genotyping of HR-HPV is becoming increasingly common in HPV-based screening; this is differentiated identification of the most oncogenic genotypes, which implies a more conservative approach to the management of patients in whom other, less aggressive genotypes are found [25]. Nevertheless, recent studies indicate that (a) “more aggressive” HPV genotypes do not significantly differ from the less aggressive ones in the probability and rate of clearance [26], and (b) their contribution to the incidence of cervical disease may gradually decline as a consequence of vaccination programs targeting more aggressive types. Our data also do not support the concept of partial genotyping because the relative frequency of HPV16 and HPV18 in HPV-positive patients did not differ between any groups. Lately, alternative approaches to patient stratification by lesion risk have been developing that are based on different molecular biomarkers [27,28,29], all of which are involved in numerous causal relations with the activity of the local microbiome.

The results of our work (among many others) indicate that CVM composition reflects the severity of cervical disease. Nevertheless, to date, no algorithms have been devised for the effective use of CVM-analysis-based testing for diagnosing and assessing the risk of a cervical pathology. This situation is due to many constraints. Cervicovaginal microbial communities are very dynamic [30] and exhibit high variation related to ethnic, geographical, and individual genetic factors [17]; in fact, they are a superposition of cervical and vaginal microbiomes, which show some differences in composition [31]. These communities are not homogenous, and results of their analysis may depend on the technique of sampling [32]; they can be affected by many mechanisms, which are not necessarily linked with carcinogenesis. Therefore, their structure cannot and should not be regarded as a direct indicator of pathology. At the same time, at present, it can be regarded as proven that dysbiotic profiles of the CVM underlie an elevated risk of SIL and CC. High biodiversity of the CVM causes extra production of proinflammatory cytokines and chemokines, which enhance an inflammatory response and promote HPV infection and persistence [33,34]. Aberrant CST structure is supported by the immunosuppressive and hypoxic microenvironment triggered by the viral infection [35,36,37,38,39]. The geographic regions—where major ethnic groups have more common non-LB predominance and lower abundance of the most favorable CSTs—tend to have higher HPV infection and cancer prevalence rates [34,40]. The detection of HR-HPV during LB deficiency (and/or enrichment with the environmental species that are rarely detectable in healthy subjects) indicates a higher risk of a cervical lesion; therefore, greater attention to such patients under further observation seems reasonable (as compared to HPV-infected patients with more favorable CSTs). Moreover, because there is currently no etiotropic treatment of HR-HPV infection, microbiome correction with the help of LB-based probiotics may reduce the risk of the disease by creating conditions unfavorable for persistence of the virus. A significant positive effect of probiotics on the reversion of low-grade lesions and on HPV clearance has been demonstrated in some recent studies [41,42,43]. In this context, it is advisable to prescribe probiotics not “by default” but based on results of an analysis of the cervical microbiome.

Our study focused on evaluating CVM biomarkers in terms of potential inclusion into high-throughput diagnostic tests for use in combination with HPV testing. Although next-generation sequencing yields much more data, this methodology is currently problematic for routine analysis owing to its high cost and low throughput. Unlike microbiomes at many other localizations, a healthy cervicovaginal microbiome is characterized by relatively lower diversity and number of dominant species; therefore, its condition can be analyzed using a limited set of marker species. Largely for this reason, a number of molecular diagnostic kits have been developed and successfully introduced into clinical practice to characterize vaginal dysbiosis [14]. These kits provide information on the relative proportion of dominant species of microorganisms in the total bacterial community. This parameter for each of these species can also be derived from data obtained by 16S rRNA sequencing, although the latter does not always allow the species of a bacterium to be determined correctly [44]. When properly optimized, qPCR may have the accuracy of bacterial species quantification similar to next-generation sequencing-based methods [45], although the causes of biases for different methods vary. The 16S rRNA gene is a widely used target for PCR-based diagnostic kits because (a) the length of this gene and the level of its genetic variation usually allows for accurate determination of the bacterium species by PCR, although biased results related to similarities between rRNA sequences of closely related bacterial species could not be ruled out; (b) any other gene may be absent from a given strain of any species or be replaced by a homologous gene borrowed from another species by horizontal transfer; and (c) for this gene, the number of sequences in databases is the largest. In this regard, we consider 16S rRNA sequencing data a direct source of new promising microbial biomarkers for subsequent clinical validation as components of the simpler PCR-based tests. These tests may be aimed at stratifying patients by cervical disease risks or the risks of recurrence on the basis of the bacterial-community type or clinically significant concentration of selected species.

In our view, stratification of patients by five “classic” CSTs as a way of interpreting a PCR test result may be unproductive. Particularly, in our study, *L. gasseri* and *L. jensenii* as the basis for the two of three “favorable” CSTs (II and IV) showed low abundance even in normal cytology samples, whereas *L. iners* (a key component of CST III) dominated very frequently in all groups. This finding is fully consistent with our previous results about Novosibirsk and Moscow populations (unpublished data) and a recent large-scale study from China [40]. Undoubtedly, *L. iners* is less protective than other LBs [46], and its dominance is conducive to transition from a healthy microbial community to an LB-deficient microbial community; such a transition can be triggered by age-related physiological changes, a comorbidity, or treatment. Accordingly, quantitative assays of *L. iners* may be clinically useful for managing the risk of cervical-disease onset and/or recurrence. On the other hand, widespread high prevalence of *L. iners* may reduce the predictive value of the assay when *L. iners* is used as a biomarker.

As for the main opportunistic species whose concentration rises significantly with increasing lesion severity, they can be regarded as acceptable biomarkers of dysbiosis to be included in diagnostic tests. Despite some known regional and ethnic differences in major cervicovaginal CST variants, a relatively small set of such species can be employed to detect most cases of anaerobic and aerobic dysbiosis. In our laboratory, the results of the analysis by a commercial qPCR-based kit aimed at detecting BV and AV obviously correlated with the risk of cervical lesion. In this regard, since BV and AV are themselves proven risk factors for cervical disease, we wonder whether it makes sense to upgrade existing diagnostic tools to detect these dysbioses in order to more reliably stratify patients by cervical pathology risk or to apply them “as is.” When answering this question, in our opinion, one should keep in mind that severe cervical lesions are more likely to be found in women at the age characterized by age-related hormonal changes affecting the microbiome and shifting its composition toward the prevalence of species characteristic of BV and AV, regardless of the presence of intraepithelial neoplasia [47,48]. Therefore, we believe that a laboratory test for stratification by risk of cervical pathology, and even more so its recurrence, should include additional markers more specifically associated with it (or its absence). In our work, most of the typical bacterial markers of BV and AV worked well at discriminating the elevated high-grade lesion risk, as confirmed by both next-generation sequencing- and PCR-based methods, but were found to have rather high relative abundance in the PT group. Some species, however, in particular *Cutibacterium acnes*, *Ralstonia pickettii*, and *Achromobacter* species, did not follow this pattern. A reduction in their relative abundance can be regarded as a sign of post-treatment microbiome recovery even against the background of LB deficiency. All these species are aerobic and are often extracted from a wide range of environmental niches (e.g., soils and water) but are considered opportunistic human pathogens accompanying inflammatory processes. For instance, *Ralstonia* is a common cause of intrahospital infections in immunocompromised patients [49]. *Achromobacter* species may be components of normal intestinal microflora in healthy individuals but have been identified as opportunistic pathogens in immunosuppressive conditions such as cystic fibrosis, cancer, and renal failure [50]. *C. acnes*, a representative of the normal skin microbiota, is one the most abundant microorganisms in healthy adults. Nonetheless, it is widely known for its association with folliculitis and acne and can cause chronic blepharitis, endophthalmitis, and endocarditis owing to inflammation induction and biofilm-forming potential. In addition, *C. acnes* can persist on implants and surgical devices, thereby causing postoperative infections [51]. It is possible that these species can actively colonize the cervical epithelium only in case of a substantial impairment of local immunity. A molecular test that includes quantitation of these species (or species having similar enrichment patterns) may help to assess the risk of cervical-disease recurrence and the need to correct the microbiome after the treatment, although large clinical studies are needed to validate such an approach.

Interpretation of the results of microbiome analysis in PT patients deserves special attention. There are still few data on this subject. It has been reported that immediately after local removal of a cervical neoplasia, CVM composition tends to shift toward a normal range of parameters during several months and even within a year [33,52]. On the other hand, in a pilot study on a small group of patients, cervical microbiome biodiversity did not decrease immediately after CRT [53]. In our study, the long-term effects on microbiome composition were found to depend on the treatment type; usually, after treatment, this composition shifted toward higher biodiversity with enrichment not only in species typical of cervicovaginal communities but also in skin-borne and environmental bacteria, intense colonization by which indicates considerable suppression of local immunity. Treatment strategies involving surgery could have led to more effective microbiome restoration in comparison with RT and especially CRT. This finding may be partially explained by greater cell damage, inflammatory processes, and programmed cell death resulting from these types of medical intervention. Furthermore, patients with a more localized lesion (which allows them to avoid adjunctive chemotherapy and RT) probably have a more favorable state of immunity; the latter helps to normalize the microflora. At the same time, this notion contradicts the finding that samples from patients after CT corresponded to the state of the microbiome that is closest to healthy judging by many parameters.

Our study has a number of limitations, which could have biased the results. These limitations include small sizes of the patient groups; incomplete demographic characteristics (for PT patients); lack of data obtained immediately before and after the start of treatment; and an age difference between some groups. We also cannot exclude the biases associated with the entry of some PT patients into menopause (because their age was ~50 and they did not use hormonal correction) and the resulting increase in abundance of anaerobic species and an LB deficiency due to a lack of estrogen [54]. There are no data on hormonal status of the patients under study; however, subgroups based on the treatment type did not differ significantly in mean age, and the mean age of the PT group was close to that of CC patients. At the same time, we cannot rule out the possibility that the observed differences in the composition of microbial communities are mediated by the effects of the treatment itself on hormonal status. In our study population, only patients with the first stage of cancer underwent operative treatment without adjunctive RT; therefore, we cannot exclude the biases associated with a more favorable state of the microbiome in patients of this group before treatment. This favorable state could have contributed to more effective recovery of the microbiome. Meanwhile, neither this nor earlier work in our lab revealed statistically significant differences in microbiome structure at different stages of invasive cancer. We also found no evidence of such differences in the literature.

It should be taken into account that in our work, a statistically significant difference between the groups was found only for highly abundant species. Unfortunately, current knowledge about ethnic and regional CVM variation does not allow us to rule out that other species may be identified as the best theranostic biomarkers (based on a microbial community) for cervical disease when other populations are analyzed and other next-generation sequencing-based assays are used. Our data were obtained in this geographic region (Western Siberia) for the first time and may complement the picture of possible CVM variations associated with cervical pathologies.

## 5. Conclusions

We demonstrated that CVM composition reflects the severity of cervical pathology. LB deficiency, high biodiversity, and enrichment with opportunistic species may be considered biomarkers of dysbiosis and an elevated risk of cervical disease. A medical intervention into cervical cancer may cause long-term microbiome changes and high biodiversity several years after the treatment. Different types of treatment differ in their effects on microbial communities. Post-treatment microbiome alteration is accompanied by elimination of some skin-borne and environmental species but not the well-known markers of women’s urogenital dysbioses. Relatively simple and high-throughput PCR tests can provide valuable information about the CVM state and may be suitable for algorithms of cervical-cancer prevention. To benefit more from knowledge about the microbiome state of a patient, further expanded research is needed. This research requires expanded application of high-throughput microbial-marker-based diagnostic tests in cervical-cancer prevention and management.

## Figures and Tables

**Figure 1 diagnostics-13-00140-f001:**
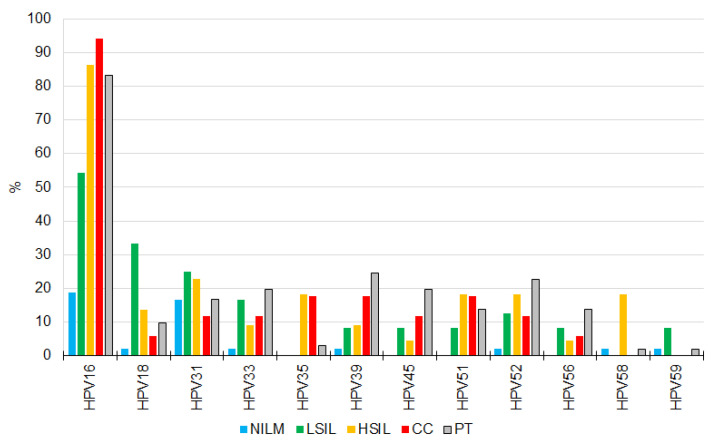
Prevalence (%) of 12 HR-HPV genotypes at different diagnoses and post-treatment. Red, cervical cancer; orange, HSIL; green, LSIL; blue, NILM; gray, the post-treatment (PT) group.

**Figure 2 diagnostics-13-00140-f002:**
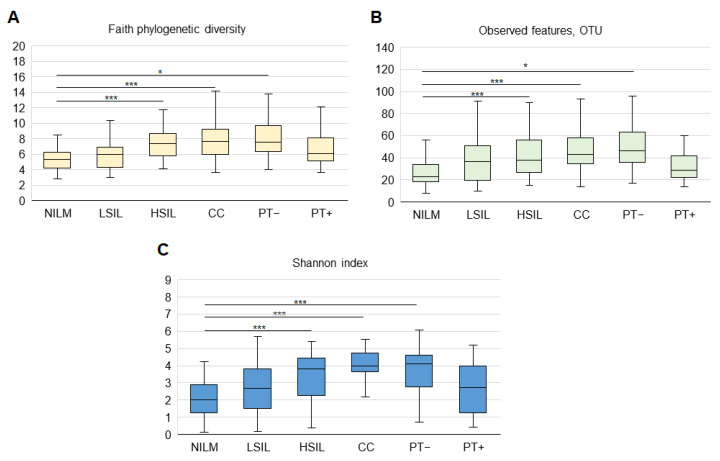
Box–whisker plots for CVM α-diversity parameters at different diagnoses and post-treatment. Box cross-sections, medians; boxes, upper and lower quartiles; whiskers, nonoutlier ranges. Outliers are not shown. PT+, post-treatment, treatment included surgery; PT-, post-treatment, treatment without surgery. (**A**) Faith’s phylogenetic diversity; (**B**) Shannon’s index; (**C**) Operational taxonomic units (OTUs). Differences between groups were assessed by the Mann–Whitney U test with Bonferroni’s correction for multiple testing, and an adjusted *p*-value was calculated as a pairwise nonadjusted *p*-value multiplied by the total number of comparisons. *** *p* ≤ 0.001; * 0.01 < *p* < 0.05.

**Figure 3 diagnostics-13-00140-f003:**
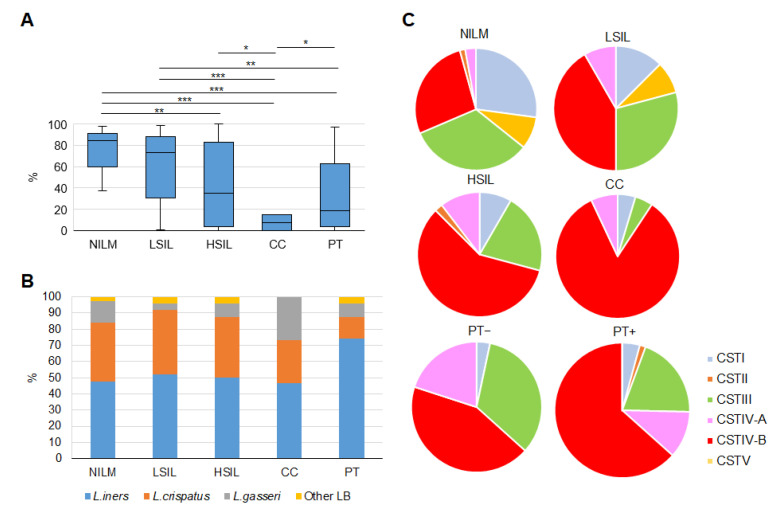
LB species analysis by 16S rRNA gene sequencing. (**A**) Enrichment with LBs (relative proportion of reads mapped to genomes of *Lactobacillus* spp.) at different diagnoses and post-treatment. Differences between groups were assessed by the Mann–Whitney *U* test with Bonferroni’s correction for multiple testing, and an adjusted *p*-value was calculated as a pairwise nonadjusted *p*-value multiplied by the total number of comparisons. The figure presents the median value, upper and lower quartiles, and a nonoutlier range; outliers are not shown. *** *p* ≤ 0.001, ** 0.001 < *p* < 0.01, * *p* < 0.05. (**B**) Relative dominance frequency of different LB species (read count > 50% of the total LB-mapped reads) at different diagnoses and post-treatment. (**C**) Distribution of microbial-community state types based on predominant LB species (CSTs I–V) among the groups of patients.

**Figure 4 diagnostics-13-00140-f004:**
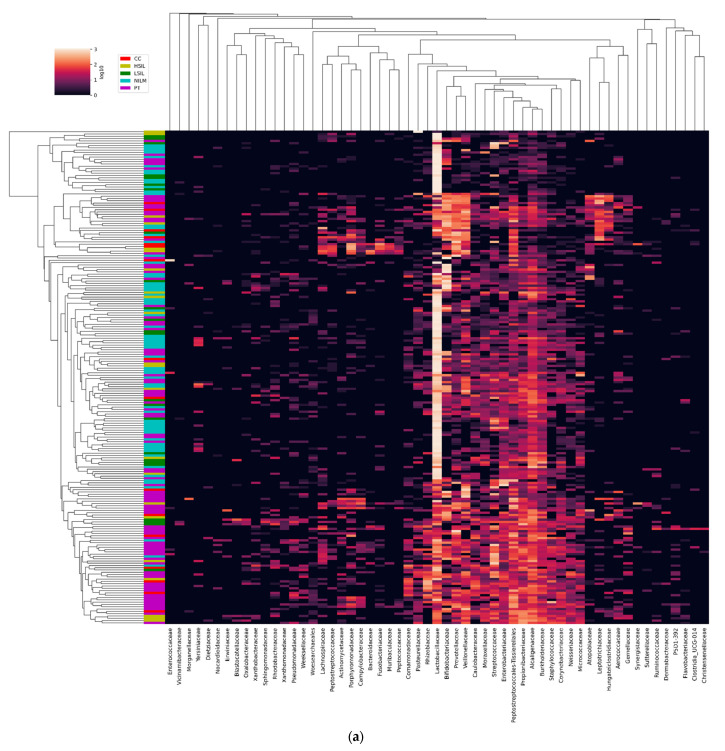
Community composition of cervical samples at the level of families (**a**) or species (**b**) as determined by massively parallel sequencing on the MiSeq platform. An unsupervised heatmap of the relative abundance of microbial taxa found in the cervicovaginal microbial communities of 234 patients: 72 NILMs (5 samples were excluded for technical reasons), 24 LSILs, 22 HSILs (2 samples were excluded for technical reasons), 15 CCs (1 sample was excluded for technical reasons), 101 PT samples, based on the Bray–Curtis dissimilarity metric. Diagnoses of patients are indicated by colors. (**a**) The families present in relative abundance of 1% in at least one sample are listed on the X axis. (**b**) Top 62 species (by abundance) are listed on the X axis. The cladograms at the top of the species’ and families’ names indicate the approximate evolutionary relationships between the species. LB species analysis by 16S metagenome sequencing. (**a**) The enrichment with lactobacilli (relative proportion of reads mapped to genomes of *Lactobacillus* spp.) at different diagnoses and post-treatment.

**Figure 5 diagnostics-13-00140-f005:**
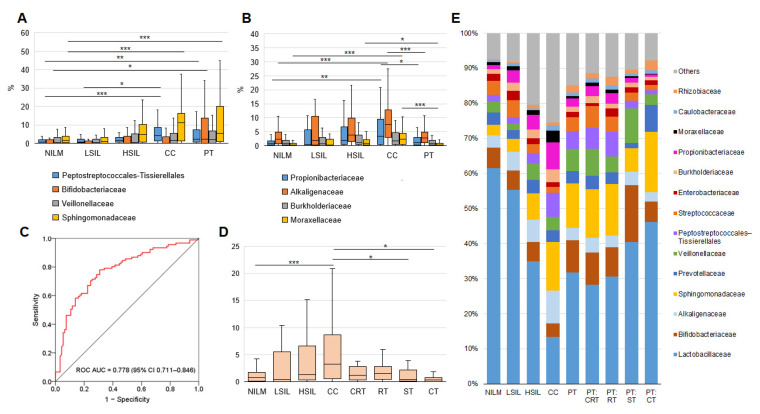
Non-LB species analysis by 16S rRNA gene sequencing. (**A**) Relative abundance counts of Peptococcales-Tissierellales, Bifidobacteriaceae, Veillonellaceae, and Sphingomonadaceae at different diagnoses and post-treatment. (**B**) Relative abundance counts of Propionibacteriaceae, Alcaligenaceae, Burkholderiaceae, and Moraxellaceae at different diagnoses and post-treatment. (**C**) The receiver-operating characteristic curve derived from training sample analysis for the detection of ≥HSIL based on non-LB reads’ % count. The diagonal line indicates an area under the curve of 0.5. (**D**) Relative abundance counts of *Cutibacterium acnes* at different diagnoses and post-treatment. Differences between groups were assessed by the Mann–Whitney U test with the Bonferroni correction for multiple testing; an adjusted *p*-value was calculated as a pairwise nonadjusted *p*-value multiplied by the total number of comparisons. The figure presents the median value, upper and lower quartiles, and a nonoutlier range; outliers are not shown. *** *p* ≤ 0.001; ** 0.001 < *p* < 0.01; * *p* < 0.05. (**E**) Community compositions at different diagnoses and in post-treatment subgroups. A bar chart of relative abundance of species per group. CRT, chemoradiotherapy; RT, radiotherapy; ST, surgical treatment; CT, combined therapy; PT, post-treatment group.

**Figure 6 diagnostics-13-00140-f006:**
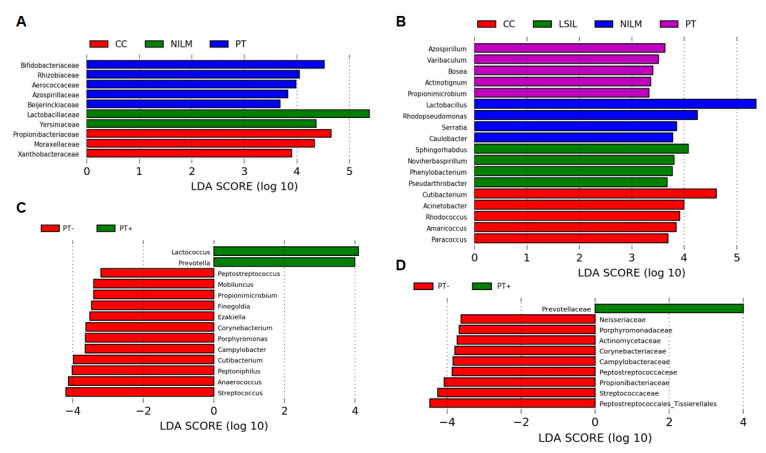
Histograms of linear discriminant analysis (LDA) scores computed for features differentially abundant among patient groups at the family and genus levels according to diagnosis (**A**,**B**) and treatment type (**C**,**D**). The LDA score on the log_10_ scale is indicated at the bottom. The greater the LDA score, the more significant the phylotype biomarker is in the comparison.

**Figure 7 diagnostics-13-00140-f007:**
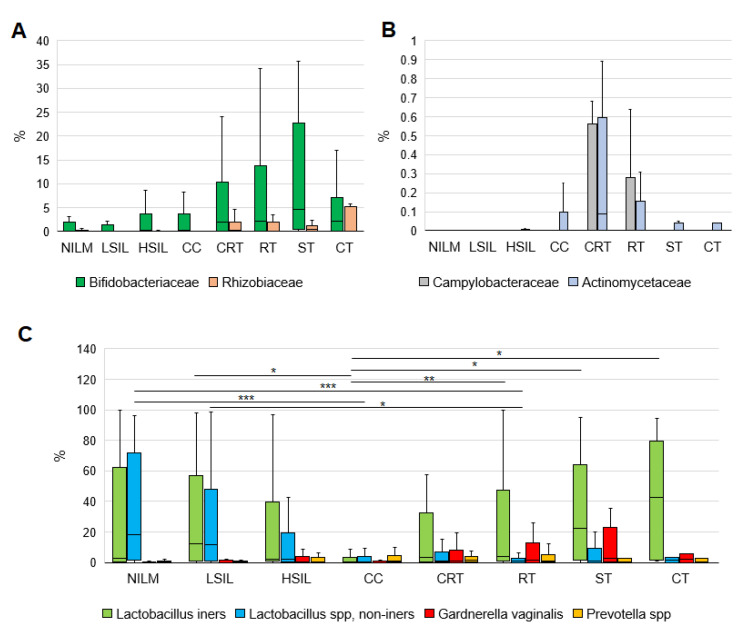
(**A**) Relative abundance counts of the highly abundant families that were found by LDA to be most significantly over-represented in the PT group. (**B**) Relative abundance counts of the low-abundance families that were found by LDA to be significantly over-represented in the PT+ vs. PT- group. CRT, chemoradiotherapy; RT, radiotherapy; ST, surgical treatment; CT, combined therapy. (**C**) Enrichment with *Lactobacillus iners*, non-iners lactobacilli, *Gardnerella vaginalis*, and *Prevotella* species at different diagnoses and in PT subgroups. Differences between groups were assessed by the Mann–Whitney U test with the Bonferroni correction for multiple testing; an adjusted *p*-value was calculated as a pairwise nonadjusted *p*-value multiplied by the total number of comparisons. The figure presents the median value, upper and lower quartiles, and a nonoutlier range; outliers are not shown. *** *p* ≤ 0.001; ** 0.001 < *p* < 0.01; * *p* < 0.05.

**Figure 8 diagnostics-13-00140-f008:**
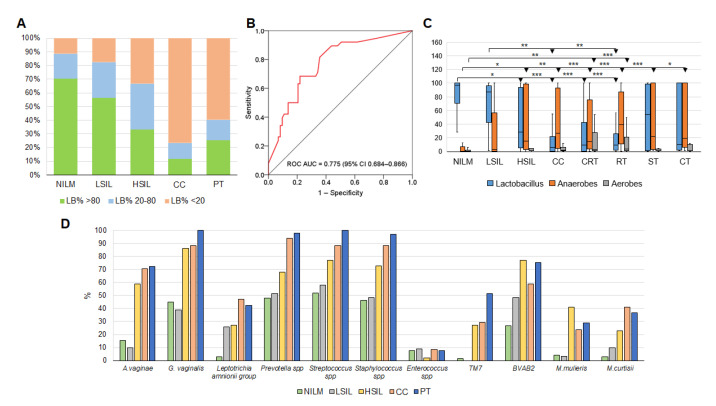
(**A**) At different diagnoses and post-treatment, stratification of cytological specimens by LB proportion based on BioFlor Kit analysis results. (**B**) The receiver-operating characteristic curve derived from training sample analysis for the detection of ≥HSIL based on the non-LB DNA percentage assessed by the BioFlor Kit. (**C**) Relative abundance of lactobacilli, common BV-associated anaerobes (*Gardnerella vaginalis*, *Atopobium vaginae*, *Prevotella* spp., and the *Leptotrichia amnionii* group summarized) and common AV-associated aerobes (*Streptococcus* spp., *Staphylococcus* spp., and *Enterococcus* spp. summarized) assessed by the BioFlor Kit at different diagnoses and in different PT subgroups. CRT, chemoradiotherapy; RT, radiotherapy; ST, surgical treatment; CT, combined therapy. Differences between groups were evaluated by the Mann–Whitney *U* test with Bonferroni’s correction for multiple testing; an adjusted *p*-value was calculated as a pairwise nonadjusted *p*-value multiplied by the total number of comparisons. The figure presents the median value, upper and lower quartiles, and a nonoutlier range; outliers are not shown. *** *p* ≤ 0.001; ** 0.001 < *p* < 0.01; * *p* < 0.05. (**D**) Frequency of detection of different opportunistic bacteria (as a percentage of the total number of analyzed samples) at different diagnoses and post-treatment using the BioFlor Kit, RealBest Mobiluncus mulieris/Mobiluncus curtisii DNA Kit, and RealBest Saccharimonas aalborgensis (TM7)/BVAB2 DNA Kit.

## Data Availability

Raw data from this study are available upon request from the corresponding author.

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
