# Peer review of "Cervicovaginal-Microbiome Analysis by 16S Sequencing and Real-Time PCR in Patients from Novosibirsk (Russia) with Cervical Lesions and Several Years after Cancer Treatment"

_diagnostics, 2023, doi:10.3390/diagnostics13010140_

Round 1
Reviewer 1 Report
The major concern of the manuscript is the poor explanations and misunderstandings to define a microbiome study. It would appear that qPCR and 16rRNA sequencing might not be appropriate to compare for microbiome study. Because the context of the procedure and application purposes are not in a similar way. In addition, 16s metagenomic is not the correct term to explain 16srRNA sequencing. For example, lines, 63-67,....
The whole manuscript needs to be rewritten, revised, and reorganized. Moreover, the discussion part is not clear as well.
- The quality of most pictures particularly the microorganisms' names should be improved.
Author Response
Comments and Suggestions for Authors, Reviewer 1.
>The major concern of the manuscript is the poor explanations and misunderstandings to define a microbiome study.
We apologize for the lack of clarity. We have substantially revised the whole text and had the manuscript checked by an editing company.
>It would appear that qPCR and 16rRNA sequencing might not be appropriate to compare for microbiome study. Because the context of the procedure and application purposes are not in a similar way.
Indeed, this comparison of two technically very different methods presents an ideological problem. At the same time, at least in Russian clinical practice, diagnostic kits based on qPCR and similar to those tested in our work are used for high-throughput analysis of the microbiome state and are advertised for this purpose. Similar kits are becoming increasingly common in clinical practice worldwide (please see, for example, the review by Lamont et al. "Recent advances in cultivation-independent molecular-based techniques for the characterization of vaginal eubiosis and dysbiosis" Fac Rev, 2020). In at least two aspects, the 16S rRNA sequencing results can be compared to the qPCR results. Both methods allow to determine the relative proportion of DNA of certain taxa of bacteria in the total bacterial community (although sources of error in the application of each method may vary, see, for example, the work of Jeong et al. “A qRT-PCR Method Capable of Quantifying Specific Microorganisms Compared to NGS-Based Metagenome Profiling Data” Microorganisms 2022) as well as to detect various states of the microbiome corresponding to certain physiological states (see, for example, the work of Li et al. "Age-stratified analysis of vaginal microbiota dysbiosis and the relationship with HPV viral load in HPV-positive women" J Immunol Res, 2022), where researchers determined the types of community state of the vaginal microbiota by relying solely on qPCR data.
Some explanations of this matter have been added to the text of the manuscript (lines 481–493).
>In addition, 16s metagenomic is not the correct term to explain 16srRNA sequencing. For example, lines, 63-67,
We have removed this term from the manuscript.
>The whole manuscript needs to be rewritten, revised, and reorganized. Moreover, the discussion part is not clear as well.
Sorry for this oversight. We have substantially revised the whole text and eliminated many abbreviations to make the manuscript easier to read. At the current stage, in addition to the above-mentioned corrections, we have implemented the following:
added several references relevant to the subject area of the study;
slightly expanded the description of the methods;
revised some wordings for clarity;
simplified the Conclusions, moved one paragraph from the Conclusions to the Discussion;
added a table with calculated P-values corresponding to the data shown in the figures to the Supplementary Materials.
We hope that the Discussion is now clearer.
>The quality of most pictures particularly the microorganisms' names should be improved.
Apologies, we are enclosing higher-resolution pictures with corrected microorganism names.
Reviewer 2 Report
Typographical and english mistakes can be rectified.
140 HPV-tested women from Novosibirsk (Russia) with normal cy-25 tological findings, cervical lesions, or cancer and in 101 women who had recently undergone differ-26 ENT types of cancer treatment.
microbiome profiling may uncover diagnostic and prognostic markers for cervical-lesion manage-35 ment, and quantitative-PCR–based kits may be suitable for this purpose.
Autors have done this , novelty found first time used 16s RNA sequencing for this analysis
Author Response
Comments and Suggestions for Authors, Reviewer 2.
>Typographical and english mistakes can be rectified.
140 HPV-tested women from Novosibirsk (Russia) with normal cy-25 tological findings, cervical lesions, or cancer and in 101 women who had recently undergone differ-26 ENT types of cancer treatment.
microbiome profiling may uncover diagnostic and prognostic markers for cervical-lesion manage-35 ment, and quantitative-PCR–based kits may be suitable for this purpose.
We are sorry for these errors. We have substantially revised the whole text and had the manuscript checked by an editing company.
in addition, we have implemented the following:
added several references relevant to the subject area of the study;
slightly expanded the description of the methods;
revised some wordings for clarity;
simplified the Conclusions, moved one paragraph from the Conclusions to the Discussion;
added a table with calculated P-values corresponding to the data shown in the figures to the Supplementary Materials.
We hope that the manuscript is now clearer.
Round 2
Reviewer 1 Report
- Microbiome biomarker is not a common term and it seems to be updated and considered only by the microbiome. If we are focusing on the metabolites of the microbiome, it might use biomarkers, otherwise, it doesn't make sense.
- English language needs some corrections
- If the qPCR primers and probes are targeted at specific species, please consider them clearly in the materials and methods section and also in the table of primers/probes sequences. Otherwise, 16s detection by qPCR means all the bacteria can be detected not specific genera/species.
- Quality of images needs to be improved and the genus/species names can not be distinguished
Author Response
Comments and Suggestions for Authors, Reviewer 1.
- Microbiome biomarker is not a common term and it seems to be updated and considered only by the microbiome. If we are focusing on the metabolites of the microbiome, it might use biomarkers, otherwise, it doesn't make sense.
>We have removed this term from the manuscript.
- English language needs some corrections
>Since the text has been checked twice by a reliable editing company, we believe the language of the manuscript should be considered acceptable.
- If the qPCR primers and probes are targeted at specific species, please consider them clearly in the materials and methods section and also in the table of primers/probes sequences. Otherwise, 16s detection by qPCR means all the bacteria can be detected not specific genera/species.
>Sorry, we cannot provide primer sequences and probes that are part of commercial kits because they are proprietary. In the RealBest BioFlor Kit, RealBest Mobiluncus mulieris/Mobiluncus curtisii DNA and RealBest Saccharimonas aalborgensis (TM7)/BVAB2 DNA Kits, the targets are specific regions of the 16s rRNA gene, in the HPV genotyping kit, the targets are genotype-specific regions of the E6 HPV gene. We added a phrase with the relevant information to the ‘Materials and methods’ section (lines 142-145). We consider this as a common practice: usually the articles do not contain sequences of primers and probes that are part of commercial kits, but the target regions are often indicated.
We agree that false results related to the design of a particular PCR-based kit cannot be excluded, since similar bacterial species may have similar sequences of 16S rRNA gene. Actually, even sequencing this gene is not always successful for unambiguous species determination. Moreover, the number of copies of the 16S gene per bacterial genome can vary significantly (which can lead to biases when calculating the percentage of certain bacterial taxa, but does not reduce the correctness of comparisons between qPCR and 16s-sequencing results).
However, there are reasons to use this particular target for these purposes. First, any other gene can be absent in a particular strain of any kind of bacteria, or be replaced by a gene borrowed by horizontal transfer from another species (which we know not only from the literature, but also from our own practice). Secondly, the use of the same target gene for all species, in our opinion, makes normalization more correct. Thirdly, for this gene, the largest number of sequences in databases is present. The extent of this gene and the level of its genetic variation in most cases allows you to accurately determine the species of bacterium using PCR.
In any case, the specificity of identifying certain genera, species, and genotypes with these kits was confirmed by their developers in clinical trials, and our goal was not to confirm or refute it. In our work, we used these kits as they are, with all their possible disadvantages and limitations.
We added two formulations regarding the limitations of using the 16s gene as a target for qPCR to the manuscript (lines 474-481).
All other oligonucleotides used in our work are already shown in Table S1.
- Quality of images needs to be improved and the genus/species names can not be distinguished
>Apologies, we are enclosing higher-resolution pictures with corrected microorganism names. Figure S1 and Figure S2 have been changed from the previous update. For the most complex picture (Figure S1):
the font of horizontal signatures is enlarged to the maximum possible without overlap;
png files saved at 600 dpi (was 300 dpi);
added copies of files in vector graphics format (svg), whose image quality does not depend on the resolution.
Figure S1 we placed in Supplementary Materials rather than in the main body of the article because it contains a lot of information and there is no way to make the signatures of microorganism names distinguishable in A4 format. However, we believe that the readers might be interested to familiarize themselves with the information presented in this figure, since it, in fact, presents the results of 16s-sequencing as condensed as possible, on the basis of which the conclusions presented in the article are drawn.
